# Changes in Alcohol Consumption during the COVID-19 Pandemic: Evidence from Wisconsin

**DOI:** 10.3390/ijerph20075301

**Published:** 2023-03-29

**Authors:** Rachel Pomazal, Kristen M. C. Malecki, Laura McCulley, Noah Stafford, Mikayla Schowalter, Amy Schultz

**Affiliations:** 1Department of Population Health Sciences, University of Wisconsin-Madison School of Medicine and Public Health, Madison, WI 53705, USA; 2Division of Environmental and Occupational Health Sciences, University of Illinois at Chicago School of Public Health, Chicago, IL 60612, USA; 3Survey of the Health of Wisconsin, University of Wisconsin-Madison School of Medicine and Public Health, Madison, WI 53705, USA

**Keywords:** alcohol consumption, COVID-19, statewide sample

## Abstract

Alcohol consumption often increases in times of stress such as disease outbreaks. Wisconsin has historically ranked as one of the heaviest drinking states in the United States with a persistent drinking culture. Few studies have documented the impact of the COVID-19 pandemic on alcohol consumption after the first few months of the pandemic. The primary aim of this study is to identify factors related to changes in drinking at three timepoints during the first eighteen months of the pandemic. Survey data was collected from May to June 2020 (Wave 1), from January to February 2021 (Wave 2), and in June 2021 (Wave 3) among past participants of the Survey of the Health of Wisconsin. Study participants included 1290, 1868, and 1827 participants in each survey wave, respectively. Participants were asked how their alcohol consumption changed in each wave. Being younger, having anxiety, a bachelor’s degree or higher, having higher income, working remotely, and children in the home were significantly associated with increased drinking in all waves. Using logistic regression modeling, younger age was the most important predictor of increased alcohol consumption in each wave. Young adults in Wisconsin may be at higher risk for heavy drinking as these participants were more likely to increase alcohol use in all three surveys.

## 1. Introduction

Excessive alcohol consumption is linked to numerous adverse health outcomes including cancer, liver, obesity, and kidney disease, and has been shown to be linked to psychological distress and trauma [1,2,3,4,5,6,7]. Early in the COVID-19 pandemic, social restrictions effectively reduced viral transmission; however, they also introduced a host of new risks including changes in personal anxiety and stress due to social isolation [8], employment, and other economic changes [9]. Previous research also shows that social isolation [10] and stress [11,12] are important psychological factors that often predict disordered drinking. Substance use, including alcohol consumption, is used as a coping mechanism [13,14] which is exacerbated during natural disasters, pandemics, and similar high stress or traumatic experiences [15,16,17,18,19]. Thus, an increased understanding of how alcohol patterns and behaviors changed from May 2020 through August 2021 would offer important insights into how pandemics may influence these behaviors.

Early in the pandemic, surveillance focused largely on identifying case counts and were less focused on the social and behavioral impacts of lockdowns in the United States. Lockdowns, such as those implemented in early 2020 in the United States, were unprecedented over the last century. Thus, very little information exists that is temporally and culturally relevant to the US population. A survey of adults living in Hong Kong during the 2003 SARS-CoV outbreak found that 6.8% of randomly-sampled adults, and 6% of hospital employees reported increased alcohol use as a coping strategy [17,20]. An early study conducted via a survey during the SARS-CoV-2 pandemic lockdowns in China, found increased levels of depression and anxiety in a snowball sample of the general public via university students [21]. Other studies showed a greater increase in alcohol consumption as a coping strategy during the initial lockdown phase of the pandemic [22,23,24], consistent with previous findings on the psychological impact of pandemic-related quarantines [25]. In the US, alcohol sales in early March to mid-April 2020 rose significantly, with an increase in liquor store sales of 54% and online alcohol sales of 262%, compared with 2019 data [26]. Data has shown that national trends in alcohol consumption did increase across the US after the lockdowns. However, little data is available on changing trends in alcohol consumption at a state level in places like Wisconsin, an upper Mid-Western state of the United States.

In Wisconsin, alcohol overconsumption is a persistent public health burden. Wisconsin is consistently identified as one of the heaviest drinking states in the US, and has an adult population that is more likely to drink alcohol (64%) than the national average (55%). It reports the highest rate of binge drinking, defined as consuming more than four drinks for women or five drinks for men on one occasion, with 21.9% of Wisconsinites reporting binge drinking in the past month [27,28]. Wisconsin ranks third in the nation for number of adults who report drinking any alcohol, with only Washington D.C. and North Dakota reporting higher percentages [27]. Wisconsinites were less likely (38%) than the national average (45%) to perceive significant risk from weekly binge drinking [28]. Given this baseline of high drinking, and low perception of alcohol consumption risks as part of the culture, public health officials in Wisconsin were weary of an additional spike in alcohol use in response to stress incurred due to the COVID-19 pandemic. Additionally, Wisconsin was one of the only places in the United States where restrictions related to COVID-19 were placed at the county level instead of the state level. Therefore, Wisconsin is a unique space in which to study these dynamics. 

The primary aim of this study is to identify factors related to changes in alcohol use at three distinct timepoints during the COVID-19 pandemic in a statewide sample of Wisconsin.

## 2. Materials and Methods

### 2.1. The Survey of the Health of Wisconsin (SHOW) 

The Survey of the Health of Wisconsin (SHOW) is a statewide population-based health examination study which began in 2008. Data were collected to support ongoing population-based monitoring and to support innovative translational research. Data collection included survey- and exam-based measurements to address a broad range of social determinants and health outcomes [29]. The sampling frame for the SHOW COVID-19 survey data included all past SHOW adult participants (n = 5846) recruited between 2008 and 2019. More details about the SHOW cohort, sampling frame, and study design are available elsewhere [30]. 

### 2.2. The SHOW COVID-19 Community Impact Survey 

#### 2.2.1. Study Participants and Recruitment 

In spring of 2020, The SHOW developed the online COVID-19 Community Impact Survey in collaboration with over 25 professors and investigators across the University of Wisconsin, Madison. The survey was administered at three different timepoints over the course of 2020–2021 (referred to as “waves” of the survey). The survey aimed to capture COVID-19 perceptions, beliefs, and behaviors, as well as how the pandemic affected their mental, physical, and emotional health and their life overall. The online survey was administered from May through mid-July 2020 (Wave 1), January through mid-March 2021 (Wave 2), and mid-June through mid-August 2021 (Wave 3) [31]. SHOW participants were eligible to participate in any or all three waves if they had consented to be contacted for future research and have provided an email or phone number. Among the 5846 adult SHOW cohort, n = 5502 met eligibility criteria and were invited to participate in every wave of survey. 

A unique web-based survey link was emailed to all eligible participants with information on what the survey would ask. The survey was administered online via UW ICTR-CAP REDcap. Participants were also contacted by phone if they did not have a valid email address, and were asked for a valid email address at that time or had the opportunity to complete a shortened version of the survey via a phone interview. 

The study was approved by the University of Wisconsin-Madison Health Science Institutional Review Board. All participants who completed the online COVID-19 or telephone surveys received a $25 electronic gift card. 

In total, 1403 participants completed the Wave 1 survey, 1889 participants completed the Wave 2 survey, and 1854 participants completed the 1-year follow up Wave 3 survey [32]. Information on how many participants completed each survey, and how many participants completed multiple surveys, are available on the SHOW website [31]. Additionally, n = 55 completed the telephone survey. More details about the SHOW COVID-19 Community Impact Survey and the cohort and methods have been described elsewhere [32], and are available on the SHOW website [31]. 

For this study, only participants who completed the online survey and had complete data on changes in alcohol consumption were included in the analyses; those who completed the telephone interview survey were excluded. A total of n = 1290, n = 1868, and n = 1827 had complete data on alcohol consumption, and were included in analysis for waves 1, 2, and 3, respectively. 

#### 2.2.2. Alcohol Consumption Assessment 

Individuals were asked to self-report whether their alcohol consumption was “a lot more, a little more, about the same, a little lower, or much lower” in the last 60 days compared to a reference period. For each wave of data collection, the question was asked cross-sectionally. Wave I asked participants about alcohol consumption compared to before the pandemic, Wave II since 1 July 2020, and Wave III since 1 February 2021.

#### 2.2.3. Demographics and Characteristics

Gender, income, educational attainment, presence of children in the home, smoking status, remote work status, changes in employment during the COVID-19 pandemic, and anxiety and depression statuses were self-reported within the survey. Anxiety and depression statuses were determined by asking if participants had ever been told by a doctor or health care professional that they had these conditions, and that were not related to COVID-19. Self-reported race was collected in four categories, then was categorized as non-Hispanic white and non-white due to a relatively small number of non-white participants in the surveys. Age at time of survey was analyzed categorically as 21–40, 41–60, and greater than 60 years of age to group participants into relevant generational cohorts that may differ in drinking habits and in their reporting of drinking habits. Income groups were determined by self-reported annual household income less than $29,999, between $30,000 and $59,999, between $60,000 and $99,999, and greater than $100,000. Health status was assessed on a 5-point Likert scale based on the validated SF-12 health survey with possible responses being Excellent, Very Good, Good, Fair, or Poor. These were then grouped into 3 categories: Fair/Poor, Good, and Excellent/Very Good for ease of analysis. Educational attainment was grouped by High School/G.E.D. or less, Some College, and bachelor’s degree or higher to ascertain relevant cut points in average earning potential. 

#### 2.2.4. Statistical Analysis 

All statistical analyses were completed in SAS v9.4. We categorized responses related to alcohol consumption into whether participants drank more, about the same, or less than before the pandemic to ensure sufficiently large sample size in each category. Only those who completed the questions related to alcohol consumption were included in analysis. Those with incomplete demographic data were included in comparisons where they had data present and were not entirely excluded. All participants that completed a survey was included in analyses for that time point, regardless of participation in other waves. Within-survey univariate differences in changes in alcohol consumption were compared using a chi-squared test. Differences in alcohol consumption were evaluated by gender, age group, race, income, anxiety and depression status, health status, remote work status, whether the participant experienced changes in employment, and presence of children in the home. These were chosen a priori, as these factors were found to be significant in other literature.

Following univariate comparisons, we completed stepwise logistic regression modeling to model odds of increased drinking to understand how alcohol consumption changed after adjusting for other factors that were statistically significant in all three waves. Each wave was modeled separately, as there may be different factors contributing to increased drinking behavior and different phases of the COVID-19 pandemic. Participant age, modeled as a cubic spline, was the primary predictor due to nonlinear effects observed in univariate comparisons. Data was analyzed separately for each survey wave to preserve the cross-sectional nature of the questions. Tables depicting this modeling process for each wave, and plots showing the spline effects in each wave, are available in the Appendix A. After determining an optimal final model for each wave, the sample was restricted to those ages between 21 and 60 years, and stratified by presence of children in the home, to better understand the impact of children in the home on reported changes in drinking behavior among those ages likely to be raising children. Odds ratios are reported comparing odds of increased drinking at age 55 to 5-year increases in age from ages 21 to 60. Age 55 was selected as the comparison based on the spline models in the Appendix A, and because it is nearest to the average age in the sample for each wave. 

## 3. Results

### 3.1. Demographics

In the first, second, and third surveys, n = 1290, n = 1868 and n = 1827 had complete data on alcohol consumption for this analysis, respectively. Table 1 describes the demographics of the sample, including differences in changes in alcohol consumption. All timepoints were majority non-Hispanic white, female, with a bachelor’s degree or higher. In Wave 1, 23.18% of respondents reported increased drinking; in Wave 2, 18.84% of respondents reported increased drinking; and 16.10% of respondents reported increased drinking in Wave 3. Some n = 91 of 986 participants who completed all three survey waves reported increased drinking at all timepoints, and n = 39 of these participants reported decreased drinking at all three timepoints. Figure 1 demonstrates changes in alcohol consumption over the three waves. Reports of increased drinking slightly decreased in each wave, and those reporting drinking as about the same increased in each wave, from 61.47% in the first wave to 70.4% in the third wave.

### 3.2. Univariate Comparisons of Changes in Alcohol Consumption

In all three survey timepoints, changes in alcohol consumption varied significantly with anxiety status, educational attainment, age, and presence of children in the home (Table 2). Participants reporting anxiety were more likely to report increased drinking in each wave. Those with a bachelor’s degree or higher were more likely to increase drinking in each wave, compared to those with a high school education or less, or those with some college education. Those in the oldest age group were the least likely to report an increase in drinking in all three surveys compared to their younger counterparts. Participants with children in the home were more likely to increase drinking all three surveys. Those who reported working remotely were more likely to report increased alcohol consumption compared to those who did not report working remotely in all three surveys. Finally, those in the highest income quartile were more likely to report increased drinking in all surveys compared to those in lower income quartiles. Those reporting depression were significantly more likely to report increased drinking habits in the first two surveys, but not the third survey. Participants reporting changes in employment were significantly more likely to report increased drinking at the first timepoint, but not at the second or third timepoints. White participants were more likely to report similar drinking behaviors at the first timepoint, and non-white participants were more likely to report decreased drinking behaviors at the first timepoint. However, results were similar at subsequent timepoints. Those reporting Fair/Poor health at the second timepoint were less likely to report increased drinking than those reporting better health statuses. See Table 2 for a complete list of within-survey comparisons.

### 3.3. Logistic Regression Modeling of Increased Alcohol Consumption

For all three waves, age modeled as a cubic spline was the primary predictor of increased alcohol consumption. Additionally, in Wave 1, being classified as a heavy drinker at baseline participation, experiencing employment changes due to COVID-19, and educational attainment were significant predictors of increased alcohol consumption. In Wave 2, only being classified as a heavy drinker at baseline participation, and educational attainment were significant predictors of increased alcohol consumption. Finally, in Wave 3, being classified as a heavy drinker at baseline, educational attainment, and income were significant predictors of increased alcohol consumption. Those who were classified as a heavy drinker at baseline were less likely to increase drinking in all three waves, and those with higher educational attainment were more likely to increase drinking in all three waves (Table 3). We utilized stepwise logistic regression to obtain each final model. Tables depicting this process are available in the Appendix A.

To better understand how children in the home impacted odds of increased alcohol consumption among those in the age group most likely to be raising children, we performed a stratified analysis based on the adjusted logistic regression model. We restricted the final adjusted model for each wave to those aged from 21 to 60 and stratified by whether children were present in the home, to understand how effects differed among those with and without children. In Wave 1, there were significant differences for each of the age comparisons, for all comparisons except for age 55 compared with age 21 for those with children in the home, with 55-year-olds being less likely to increase drinking, except when compared with 60-year-olds, where the effect is reversed. However, none of the comparisons for those with no children in the home were significant. In Wave 2, the only significant comparisons were between participants aged 55, and those aged 35 and 40 years, respectively, with children present in the home, where 55-year-olds remained less likely to increase drinking. No other significant comparisons were reported in Wave 2. In Wave 3, there were no significant comparisons in either stratum (Table 4).

## 4. Discussion

To our knowledge, this is the first study to examine changes in alcohol consumption during several phases of the COVID-19 pandemic in Wisconsin. Alcohol consumption trends are related to several physical and mental disorders [1,2,3,4,5,6,7] that may exacerbate issues related to COVID-19. These data are a unique contribution to the literature on this topic because by utilizing serial surveys, we can examine changes in these dynamics within a relatively short period of time. Observations at multiple timepoints throughout the pandemic at the population level are unique, as most other studies focus on changes in the first few weeks or months of the pandemic. Additionally, Wisconsin is an opportune state to study these changes during the pandemic due to its strong culture of drinking, ranking third in adult binge drinking in the United States [27]. Therefore, Wisconsinites may be at higher risk for increased drinking in stressful situations, like in a global pandemic. Statewide surveys give us a clearer picture of the impact of COVID-19 on regions and communities across the state when in-person data collection was difficult.

In univariate comparisons, we found increased drinking habits among those reporting anxiety at all three timepoints and among those reporting depression during the first two timepoints. At all three timepoints, we also found increased drinking behavior among those reporting children in the home. We also found increased drinking at all three timepoints among younger age groups and those with a bachelor’s degree or higher, those in the highest income group, and those who reported working remotely due to COVID-19. However, after adjusting for other factors, younger age was the most important factor related to increased drinking in all three waves. Older participants were much less likely to report increased drinking in each wave, which may be because they did not increase alcohol consumption, or because they were more sensitive to social desirability bias in these surveys. Higher educational attainment was also a significant predictor of increased alcohol consumption in all three waves after adjustment. This may be because those with a bachelor’s degree or higher have higher socioeconomic status on average and may be more able to access alcohol due to increased means to purchase alcohol when there are pandemic-related financial strains. Being classified as a heavy drinker at baseline participation was protective against reporting increased alcohol consumption in all three waves. This may not mean that heavy drinkers were consuming less alcohol than before the pandemic, but that their habits may have remained relatively constant at a higher level. Presence of children in the home was not a significant predictor in any wave when age was also in the model; however, in the first wave of the survey the effect of age appears to be driven by whether children were present. Age and presence of children in the home are highly correlated and may be showing effects of a similar process. Different factors were significant predictors of increased alcohol consumption in each wave, likely due to the changing dynamics of the pandemic. In the first wave, changes in employment due to COVID-19 was a significant predictor of increased alcohol consumption after adjustment, but this was not the case in the other waves. This may be because the initial economic shock caused by a change in employment led to increased alcohol consumption; however, this did not persist in later months. In the third wave, higher income was a significant predictor of increased drinking. This may be mirroring the effect of higher educational attainment on increased drinking behavior. These differences between waves demonstrate the rapidly changing social environment brought on by COVID-19 and COVID-19-related restrictions, which have thus far been understudied in the US.

It is important to mention the effect of the widespread vaccination campaign for COVID-19. Vaccines became widely available to the public in April 2021 [33], which changed how Wisconsinites interacted with the virus. This shift may help to explain the decrease in reports of higher alcohol consumption in the second and third waves. Alcohol is known to exacerbate illness; hence vaccination may have decreased the risks of drinking for vaccinated individuals [34]. Although vaccine hesitancy may have increased anxiety when COVID-19 vaccines first became available [35], Chen and the co-authors similarly found that vaccination for COVID-19 was associated with a decrease in anxiety and depression symptoms [36], which may help to explain our results. As anxiety about the pandemic waned at the population level, increases in drinking as a coping mechanism may have waned as well.

Rolland et al. similarly found increases in alcohol use among younger age groups, higher educational attainment, and current psychiatric treatment among the general population of France, which mirror our results [37]. Grossman et al. also reported increased drinking habits in those reporting children in the home among adults in the US [38]. Contrary to our results, several studies found that women were more likely to increase drinking habits [39,40], and one study found higher drinking among men [41]. However, we did not find a significant difference in changes in alcohol consumption between sexes. Both studies utilized different metrics to assess alcohol consumption from those used here (number of alcohol-using days, binge drinking, and number of drinks per drinking occasion versus self-reported changes in drinking habits), which may account for some of these differences. Additionally, these studies were conducted at only one timepoint, so it is unclear whether these results would hold had the survey been completed multiple times at different phases of the pandemic. Finally, the study by Dumas et al. was conducted in Canadian adolescents, who may have different drinking habit changes compared to adults in the US due to differing attitudes toward adolescent drinking in both countries, as well as general age group differences. Karadayian et al. found an overall decrease in alcohol consumption among Buenos Aires students, but they similarly found that those in the 25–35 age group drank more [41]. We did not include participants under the age of 21 in the present analysis, which may help to explain this difference. Sugaya and colleagues also reported higher rates of unhealthy drinking habits among those whose economic situation had deteriorated due to the pandemic [42], which mirrors what we found in the first wave where more reports of increased drinking were found among those who also reported employment changes due to COVID-19. This study began later than the present study, but because pandemic restrictions remained in place longer outside of the US, it is logical that psychological effects due to the pandemic would persist longer in these areas. Comparing these results is important from a public health perspective because understanding how different populations were impacted by the social isolation of COVID-19 may have implications for long-term public health related to alcohol consumption.

This study has several strengths and limitations that may impact the results of the survey. First, 986 participants completed the survey at all three timepoints, and 1675 participants completed at least two timepoints. This repeated participation enables us to examine changes in alcohol consumption through different phases of the pandemic in many of the same participants. Additionally, the rich survey data collected allows us to explore many important associations with changes in alcohol consumption. Since Wisconsin is such an advantageous place to study alcohol consumption, it is a particular strength of this study to have conducted this work here. A limitation of this study is the need to combine all non-white race and ethnicity groups into one, as there were insufficient responses within each race and ethnicity group to draw reliable conclusions. The wording of some questions changed slightly between timepoints of the survey, which may have impacted responses. The surveys also relied on self-reports of demographics, as well as changes in behavior over time, which are vulnerable to recall bias. Clear, objective definitions of increased or decreased drinking behaviors were not defined within the survey, which rely on participants’ interpretations of questions. The use of measurement based on changes in alcohol consumption instead of objective measurement of number of drinks is a significant limitation of this study. Additionally, when asking about potentially sensitive topics like changes in alcohol consumption, employment changes, income, health status, and diagnoses of anxiety and depression, social desirability bias may be important. Participants may misreport these factors, which may have resulted in differential misclassification to ‘healthier’ statuses. Since the survey was conducted online, SHOW participants who do not have internet access or could not complete the online survey for other reasons were not included in the data. This may skew the data, as internet access may be related to certain demographics and may be related to changes in alcohol consumption throughout the pandemic. Finally, for many, the psychological effects of the pandemic persisted beyond August 2021, and a longer study period may have demonstrated these effects. However, because nearly all COVID-19 restrictions were lifted in Wisconsin during the summer of 2021, and vaccines were widely available [33], a longer study period was difficult to justify.

Previous findings suggest that alcohol consumption and mental health are highly correlated [10,11]. More research is needed to understand the scope of alcohol and substance use changes in the various phases of the pandemic, and how this may impact public health going forward as populations continue to deal with COVID-19. Future studies should examine differences in alcohol consumption changes between pandemic phases by conducting longitudinal analyses, going beyond the within-phase comparisons here. These studies should also use rich survey data provided by SHOW to link COVID-19 Impact Survey data with other important exposures like housing, geography, residential history, and biological samples to better understand these dynamics in Wisconsin. Finally, longitudinal follow-up on the impacts of COVID-19 among these participants should be conducted as Wisconsinites change the ways in which they interact with alcohol and with the virus in the long term.

## 5. Conclusions

Our study suggests that certain groups may have been differentially at risk for increased alcohol consumption throughout of the COVID-19 pandemic, which may put them at elevated risk for adverse health outcomes. Healthcare providers should pay special attention to groups such as younger patients and those with higher educational attainment, and should consider discussing the risks of increased alcohol consumption with them if they believe that behavior is of concern.

## Figures and Tables

**Figure 1 ijerph-20-05301-f001:**
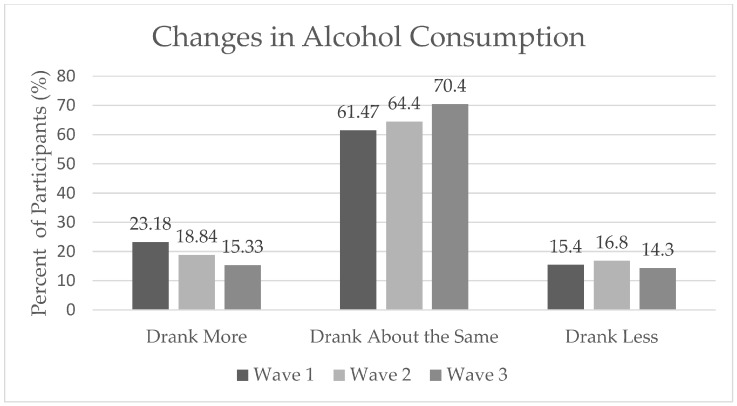
Alcohol Consumption across all Survey Waves.

**Table 1 ijerph-20-05301-t001:** Selected Demographics and Characteristics of Each Wave.

	Wave 1 (n = 1290)		Wave 2 (n = 1868)		Wave 3 (n = 1585)	
	n	Percent (%)	n	Percent (%)	n	Percent (%)
**Gender**						
Male	464	36.2	725	39.1	593	37.8
Female	817	63.8	1129	60.9	978	62.3
**Age**						
21–35 years	151	11.7	175	9.4	139	8.8
36–55 years	422	32.8	608	32.6	484	30.8
56–75 years	614	47.7	883	47.3	787	50.0
Greater than 75 years	101	7.8	201	10.8	163	10.4
**Race**						
White (Non-Hispanic)	1139	88.4	1624	87.0	1371	87.9
Non-White	149	11.6	242	13.0	189	12.1
**Education**						
H.S./G.E.D. or Less	197	15.4	301	16.2	246	15.7
Some College	411	32.0	648	34.8	555	35.4
Bachelor’s or Higher	675	52.6	912	49.0	769	49.0
**Income**						
<$30,000	163	13.9	252	15.0	208	14.7
$30,000–$59,999	301	25.6	447	26.7	362	25.5
$60,000–$99,999	347	29.5	498	29.7	417	29.4
>$100,000	364	31.0	480	28.6	431	30.4
**Self-Reported Health**						
Excellent or Very Good	783	60.7	1091	58.4	939	59.3
Good	393	30.5	609	32.6	484	30.6
Fair or Poor	113	8.8	167	8.9	160	10.1
**Children in Home**						
Children Present	379	29.4	528	28.3	401	25.3
No Children Present	911	70.6	1340	71.7	1184	74.7
**Change in Alcohol Consumption**						
Drank More	299	23.2	352	18.8	243	15.3
Drank about the Same	793	61.5	1208	64.4	1116	70.4
Drank Less	198	15.4	313	16.8	216	14.3

H.S. = High School; G.E.D. = General Educational Development.

**Table 2 ijerph-20-05301-t002:** Changes in Alcohol Consumption by Demographics.

	Wave I				Wave II				Wave III			
	Drank More (%)	Drank the Same (%)	Drank Less (%)	*p*-Trend	Drank More (%)	Drank the Same (%)	Drank Less (%)	*p*-Trend	Drank More (%)	Drank the Same (%)	Drank Less (%)	*p*-Trend
**Race**												
White	23.09	62.86	14.05	**0.0006**	18.72	64.35	16.93	0.7832	16.27	68.65	15.08	0.6943
Non-White	24.16	50.34	25.5		19.83	64.88	15.29		14.54	68.72	16.74	
**Gender**												
Male	20.91	64.66	14.44	0.1671	17.93	63.72	18.34	0.2587	14.29	69.39	16.33	0.273
Female	24.6	59.36	16.03		19.57	64.92	15.5		16.93	68.26	14.8	
**Income**												
<$29,999	17.79	58.28	23.93	**0.0001**	15.08	66.27	18.65	**0.0001**	13.28	70.12	16.6	**0.0139**
$30,000-$59,999	18.94	65.12	15.95		18.12	66.44	15.44		14.25	70.05	15.7	
$60,000-$99,999	23.92	64.84	11.24		17.87	67.87	14.26		17.49	69.55	12.96	
>$100,000	30.22	54.95	14.84		25.42	54.17	20.42		20.6	61.2	18.2	
**Anxiety Status**												
Anxiety	30.68	54.58	14.74	**0.0066**	24.93	60.98	14.09	**0.0027**	20.74	65.34	13.92	**0.0265**
No Anxiety	21.37	63.14	15.5		17.34	65.24	17.41		14.92	69.42	15.66	
**Depression Status**												
Depression	30.13	56.33	13.54	**0.0224**	23.77	60.93	15.3	**0.0261**	18.42	68.42	13.16	0.2486
No Depression	21.68	62.58	15.74		17.64	65.25	17.11		15.49	68.69	15.82	
**Remote Work Status**												
Remote Work	35.17	52.91	11.93	**<0.0001**	26.41	54.52	19.07	**<0.0001**	23.49	58.43	18.07	**0.0067**
No Remote Work	19.11	64.38	16.51		16.72	67.17	16.11		15.29	69.66	15.05	
**Health Status**												
Excellent/Very Good	23.37	62.45	14.18	0.3322	20.16	62.97	16.87	**0.0233**	16.59	67.68	15.73	0.1929
Good	24.43	59.03	16.54		19.05	65.19	15.76		15.66	67.99	16.35	
Fair/Poor	17.7	62.83	19.47		9.58	70.66	19.76		14.29	75.66	10.05	
**Education**												
HS/GED or less	15.23	67.51	17.26	**0.0016**	14.95	70.43	14.62	**<0.0001**	13.54	73.61	12.85	**<0.0001**
Some College	19.95	63.26	16.79		16.82	68.36	14.81		12.81	73.91	13.28	
Bachelor’s Degree or Higher	27.7	58.37	13.93		21.49	59.87	18.64		19.1	63.16	17.74	
**Age Group**												
21–40	35.27	48.06	16.67	**<0.0001**	26.85	56.79	16.36	**<0.0001**	22.56	58.92	18.52	**<0.0001**
41–60	28.84	56.63	14.53		23.72	59.97	16.31		20.72	64.8	14.49	
>60	12.77	71.94	15.29		12.26	70.49	17.25		10.21	74.66	15.14	
**Employment Change During COVID-19**												
Changes in Employment	25.36	59.52	15.12	**0.0387**	19	64.03	16.97	0.8673	15.24	69.4	15.36	0.6743
No Changes in Employment	19.11	65.11	15.78		18.45	65.31	16.24		16.75	67.95	15.3	
**Presence of Children in the Home**												
Children in Home	34.56	51.98	13.46	**<0.0001**	25.57	59.66	14.77	**<0.0001**	22.38	63.81	13.81	**<0.0001**
No Children in Home	18.44	65.42	16.14		16.19	66.27	17.54		13.79	70.35	15.86	

*p*-trend = *p*-value obtained from Chi-Squared test; H.S. = High School; G.E.D. = General Education Development. Bolded *p*-values indicate statistical significance at the α = 0.05 significance level.

**Table 3 ijerph-20-05301-t003:** Unadjusted and Adjusted Odds Ratios of Increased Alcohol Consumption for 5-year Age Differences in Each Wave.

	Wave I						Wave II						Wave III					
	Unadjusted			Adjusted ^a^			Unadjusted			Adjusted ^b^			Unadjusted			Adjusted ^c^		
	OR	CI Lower	CI Upper	OR	CI Lower	CI Upper	OR	CI Lower	CI Upper	OR	CI Lower	CI Upper	OR	CI Lower	CI Upper	OR	CI Lower	CI Upper
55 vs. 21 years old	0.63	0.36	1.11	0.64	0.34	1.19	0.75	0.45	1.28	0.81	0.43	1.52	0.81	0.48	1.37	0.87	0.44	1.71
55 vs. 25 years old	0.65	0.4	1.05	0.66	0.38	1.13	0.77	0.49	1.21	0.82	0.47	1.41	0.82	0.52	1.28	0.87	0.48	1.56
55 vs. 30 years old	**0.67**	0.46	0.99	0.69	0.44	1.06	0.78	0.54	1.12	0.82	0.53	1.28	0.82	0.57	1.18	0.87	0.54	1.39
55 vs. 35 years old	**0.69**	0.52	0.93	0.71	0.51	1	0.79	0.6	1.05	0.83	0.59	1.17	0.82	0.63	1.08	0.87	0.61	1.24
55 vs. 40 years old	**0.72**	0.59	0.87	**0.74**	0.59	0.94	**0.81**	0.67	0.98	0.84	0.66	1.06	**0.83**	0.69	0.99	0.87	0.68	1.11
55 vs. 45 years old	**0.75**	0.67	0.84	**0.78**	0.68	0.89	**0.83**	0.75	0.93	**0.85**	0.74	0.98	**0.84**	0.76	0.94	0.88	0.76	1.01
55 vs. 50 years old	**0.83**	0.79	0.87	**0.85**	0.8	0.9	**0.88**	0.85	0.93	**0.89**	0.84	0.95	**0.89**	0.85	0.93	**0.91**	0.86	0.97
55 vs. 60 years old	**1.35**	1.25	1.46	**1.33**	1.23	1.43	**1.23**	1.16	1.3	**1.23**	1.16	1.32	**1.21**	1.13	1.28	**1.17**	1.1	1.25

OR = Odds Ratio; CI = Confidence Interval; ^a^ Adjusted for heavy drinking at baseline participation, changes in employment due to COVID-19, and educational attainment; ^b^ Adjusted for heavy drinking at baseline participation and educational attainment; ^c^ Adjusted for heavy drinking at baseline participation, educational attainment, and income; bolded OR indicates statistical significance at the 0.05 level.

**Table 4 ijerph-20-05301-t004:** Adjusted Odds Ratios of Increased Alcohol Consumption for 5-year Age Differences in Each Wave Stratified by Presence of Children in the Home.

	Wave I						Wave II						Wave III					
	Children Present			No Children Present			Children Present			No Children Present			Children Present			No Children Present		
	OR	CI Lower	CI Upper	OR	CI Lower	CI Upper	OR	CI Lower	CI Upper	OR	CI Lower	CI Upper	OR	CI Lower	CI Upper	OR	CI Lower	CI Upper
Difference 55 vs. 21	0.32	0.09	1.16	2.13	0.50	9.06	0.46	0.12	1.76	0.43	0.12	1.53	1.57	0.36	6.88	0.70	0.17	2.85
Difference 55 vs. 25	**0.29**	0.10	0.85	1.68	0.55	5.15	0.44	0.15	1.31	0.55	0.20	1.46	1.26	0.38	4.22	0.78	0.27	2.29
Difference 55 vs. 30	**0.26**	0.10	0.62	1.25	0.59	2.66	0.42	0.17	1.00	0.74	0.38	1.45	0.95	0.37	2.46	0.89	0.43	1.84
Difference 55 vs. 35	**0.23**	0.10	0.53	0.93	0.54	1.60	**0.40**	0.17	0.91	1.01	0.60	1.70	0.72	0.31	1.67	1.02	0.59	1.75
Difference 55 vs. 40	**0.22**	0.09	0.54	0.75	0.43	1.31	**0.41**	0.17	0.97	1.25	0.71	2.20	0.58	0.24	1.38	1.11	0.63	1.94
Difference 55 vs. 45	**0.30**	0.14	0.63	0.74	0.45	1.21	0.51	0.25	1.02	1.28	0.78	2.10	0.59	0.27	1.25	1.11	0.69	1.79
Difference 55 vs. 50	**0.53**	0.35	0.79	0.84	0.64	1.11	0.70	0.48	1.02	1.15	0.88	1.49	0.74	0.49	1.12	1.06	0.82	1.36
Difference 55 vs. 60	**1.90**	1.27	2.86	1.19	0.90	1.56	1.42	0.98	2.07	0.87	0.67	1.13	1.35	0.89	2.04	0.94	0.73	1.21

OR = Odds Ratio 6; CI = Confidence Interval; bolded OR indicates statistical significance at the 0.05 level.

## Data Availability

We utilized restricted data from the Survey of the Health of Wisconsin (SHOW) COVID-19 Community Impact Surveys. Data are available upon request at show.wisc.edu. A public-use version of these data is available at no cost.

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
