# Peer review of "Changes in Alcohol Consumption during the COVID-19 Pandemic: Evidence from Wisconsin"

_ijerph, 2023, doi:10.3390/ijerph20075301_

Round 1

Reviewer 1 Report

The manuscript of Pomazan and their co-authors is well-written and shows a new data analysis on rising alcohol consumption in Wisconsin during a pandemic. The study is strong, conclusions correspond to the tasks set. There are some minor comments.

1. The authors argue that during the third wave, higher income was a significant predictor of increased alcohol consumption. The second and third waves of the pandemic (February 2021) coincided with the news about the development of new vaccines and the start of global vaccination. A few recent studies declared that vaccination and fear of vaccines approved for emergency use were significant factors of additional anxiety with following increased alcohol consumption. (10.3390/pathogens12020163 ;10.1016/j.jad.2021.10.134 ; 10.1016/j.socscimed.2022.114820).  The role of vaccination should be discussed in the manuscript.

2. It would be nice to make a couple of figures (graphs, diagrams) to make the manuscript more visual and attractive for readers.

Reviewer 2 Report

The paper is interesting and actual as it investigate the alcohol consumption during a "particular" period of the "world life".

I appreciated it and I want to suggest to the authors some little modifications that could improve the paper.

As it is a longitudinal survey to present  one or more imagines that display the evolution along the three administrations of the consumption could have more visually impact in the readers. At the same time, could be interesting to understand if the subjects that declare to "drank more" or to "drank less" are the same during the three evaluations.

Last but not least advice, search better in MDPI, because there are papers pubblished on the same topic that coud be useful to improve your discussion. 

Reviewer 3 Report

A very well-written paper, the results are interesting and are presented clearly. I have minor comments/questions for the authors:

1. It wasn't immediately clear to me why the authors chose the May 2020-August 2021 time period. Given COVID-19 pandemic and its consequences sustained well beyond August 2021, I think it'd be useful to utilize a longer time period.

2. Authors state in the introduction that Wisconsin ranks 3rd in the US for adult binge drinking- which states rank before Wisconsin?

3. I suggest the authors tidy up the tables, data presented can be clearer/easier to understand.

4. There are a few typos and grammatical errors in the manuscript, the authors need to edit it before re-submission.

Author Response

Reviewer 3:

A very well-written paper, the results are interesting and are presented clearly. I have minor comments/questions for the authors:

3.1 Reviewer Comment: It wasn't immediately clear to me why the authors chose the May 2020-August 2021 time period. Given COVID-19 pandemic and its consequences sustained well beyond August 2021, I think it'd be useful to utilize a longer time period.

Author Response: Thank you for this comment. Certainly, the pandemic and its consequences were felt long after August 2021. In Wisconsin, nearly all pandemic-related restrictions were lifted during the summer of 2021, and these restrictions have largely not returned since. Additionally, the period of study was used because the study was funded for three waves of data collection. We have included the study period as a limitation in the discussion section:

Page 11: “Finally, for many the psychological effects of the pandemic persisted beyond August 2021, and a longer study period may have demonstrated these effects. However, because nearly all COVID-19 restrictions were lifted in Wisconsin during the summer of 2021, and vaccines were widely available [33], a longer study period was difficult to justify.”

The SHOW cohort is available for investigators to continue COVID-19 research and follow-up of the cohort if additional funding is awarded in the future.

3.2 Reviewer Comment: Authors state in the introduction that Wisconsin ranks 3rd in the US for adult binge drinking- which states rank before Wisconsin?

Author Response: This was an error in the introduction – Wisconsin ranks 3rd in number of adults who report drinking (ranking behind Washington D.C. and North Dakota). Wisconsin ranks 1st in the country for adult binge drinking. This change is reflected in the introduction:

Page 2: “Wisconsin is consistently identified as one of the heaviest-drinking states in the US and has an adult population that is more likely to drink alcohol (64%) than the national av-erage (55%), and reports the highest rate of binge drinking, defined as consuming more than 4 drinks for women or 5 drinks for men on one occasion, with 21.9% of Wiscon-sinites reporting binge drinking in the past month [27,28]. Wisconsin ranks third in the nation for adult binge drinking, defined as consuming more than 4 drinks for women or 5 drinks for men on an occasion, (22% of Wisconsinites compared to 16% national av-erage)number of adults who report drinking any alcohol, with only Washington D.C. and North Dakota reporting higher percentages [2927].”

3.3 Reviewer Comment: I suggest the authors tidy up the tables, data presented can be clearer/easier to understand.

Author Response: Thank you for this comment. We have re-formatted the tables slightly such that they are hopefully easier to understand. We have also included a figure showing reported changes in alcohol consumption across waves, which we believe clarifies the results. Specific suggestions for what to tidy up to ensure they are clearly interpretable would be helpful.

3.4 Reviewer Comment: There are a few typos and grammatical errors in the manuscript, the authors need to edit it before re-submission.

Author Response: Thank you for catching these. We have checked the manuscript and corrected these errors.